# Mechanical Properties and Constitutive Model of the Cement-Improved Loess under Freeze-Thaw Conditions

**DOI:** 10.3390/ma15197042

**Published:** 2022-10-10

**Authors:** Yaqiang Niu, Luzheng Hou, Zipeng Qin, Xu Wang, Yongfu Zhang, Weilong Shao, Guangrong Jiang, Xianding Guo, Junsuo Zhang

**Affiliations:** 1School of Civil Engineering, Lanzhou Jiaotong University, Lanzhou 730070, China; 2Key Laboratory for Technology in Rural Water Management of Zhejiang Province, School of Water Conservancy and Environment Engineering, Zhejiang University of Water Resources and Electric Power, Hangzhou 310018, China; 3Shaoxing Keqiao District Water Resources Development Center, Shaoxing 312030, China; 4Cangnan County Water Conservancy Bureau of Wenzhou City, Wenzhou 325800, China

**Keywords:** cement-improved loess, freeze-thaw condition, mechanical properties, nonlinear Mohr-Coulomb strength criterion, binary-medium model

## Abstract

Cement-improved loess (CIL) is used as a common filler for subgrade construction projects in loess areas. The freeze-thaw (F-T) conditions have a significant effect on the stability of cement-improved loess subgrades in seasonally frozen regions. In this paper, the CIL samples, experiencing different numbers of F-T cycles at varying freezing temperatures, were used in consolidated undrained triaxial compression tests to investigate the effect of F-T conditions on the mechanical properties of CIL. The results show the stress-strain curves of CIL are of a strain-softening type with strong elastic brittleness. The initial tangent modulus of CIL increases with the growing confining pressure and gradually decreases with the increase in the F-T cycle number and the decreasing freezing temperature. It loses 46.4% of its original value after the twelfth F-T cycle with the confining pressure of 150 kPa and at the freezing temperature of −15 °C. The strength of CIL decreases with the increasing F-T cycle number, but it gradually tends to keep stable after the sixth F-T cycle. The strength also decreases with the reduction in the freezing temperature. It loses 37.7% of its original value after the twelfth F-T cycle with the confining pressure of 150 kPa and the freezing temperature of −15 °C. To express the nonlinearity correlation between the strength and confining pressure under F-T conditions, the Weibull function was applied and a nonlinear Mohr-Coulomb strength criterion was proposed. Through introducing a breakage rate function and a local strain coefficient, a binary-medium constitutive model consisting of bonded elements (soil-particle cohesion) and frictional elements (soil particles or soil aggregations) was established to describe the stress-strain relationships of CIL under F-T conditions. The test results indicated that the model can well describe the strain-softening phenomenon of the stress-strain curve of CIL and reflect the breakage mechanism of CIL.

## 1. Introduction

In recent years, the infrastructure in northwestern China, especially the high-speed railways and expressways, has witnessed a growing development. Improved loess is usually used as the subgrade filler in the regions because of the extensive distribution of loess and a lack of high-quality filler [1,2,3]. At present, various improvement methods of the loess have been researched such as traditional inorganic, ion, organic polymer, and biological curing agents [4,5,6,7,8,9,10,11,12,13]. There are also many studies on the reuse of solid waste which can be used in material improvement [14,15]. The traditional inorganic materials such as cement show stronger engineering applicability in terms of their physical and mechanical properties [16]. Therefore, they have been widely used in high-speed railways, expressways and other civil engineering projects.

The initial moisture content of cement-improved loess (CIL) that is used as a subgrade filler generally covers the range from 11% to 16% (the optimal moisture content). Water often seeps into the subgrade and is difficult to be completely drained out in the case of heavy rainfall, groundwater activity, or lateral seepage of the foundation in cloudy and humid mountainous areas. In a seasonally frozen region, at least one round of F-T cycle occurs inside the CIL subgrade per year, which will obviously alter the particle arrangement, structural connection, permeability, plasticity index and other physical properties of the soil [17,18,19,20,21,22]. It will also lead to a decrease in the strength and deformation resistance of the soil and a lower stability of the deformation and strength of CIL subgrades [23,24,25,26,27,28,29,30,31,32]. Regarding the effect of F-T conditions on the mechanical properties of improved soil, Bozbey [33] investigated the influence of the degree of pulverization on lime-improved clay. Zhang [34] and Nguyen [35] evaluated the effects of cement and lime on the frost resistance of loess. Wang [36] found that the stress-strain relationships of lime-improved and cement-improved soils are respectively the strain-hardening and strain-softening types under the condition of F-T cycles, and the cohesion of the improved soils decreases with the growing F-T cycle number. Through introducing impacting factors, Zhang [37] studied the variation law and expression of the yield surface of cinder-improved soils after the process of F-T cycles and established an elastic-plastic constitutive model. Lv [38] evaluated the changes in physical and mechanical properties of the unconfined compressive strength, permeability coefficient, and micro-structure of improved loess with different cement, lime and fly-ash ratios before and after F-T cycles.

Numerous studies on the effects of F-T conditions on strength, elastic modulus, and stress-strain relationships of different kinds of geomaterials, such as soil, concrete and asphalt binder, have been carried out [39,40]. But for CIL, it is inadequate in mechanical degradation under F-T conditions, and the constitutive model which can reasonably describe the stress-strain relationship is still limited. Since CIL is extensively used as subgrade filler in high-speed railways and highway construction in northwest China with a typical climatic of freeze-thaw conditions, it is necessary to analyze in depth the variation in the mechanical properties of CIL and establish a constitutive model that can reliably describe the deformation behavior of CIL under F-T conditions. It is critical to calculating the additional settlement and evaluating the long-term stability of CIL subgrades in seasonally frozen soil regions.

In this paper, consolidated undrained triaxial compression tests on CIL samples which experienced F-T cycles at different freezing temperatures were performed. The effects of F-T conditions on the deformation and strength behavior were investigated. A nonlinear Mohr-Coulomb strength criterion for CIL was proposed based on the nonlinear strength envelope. According to the theory of geotechnical damage mechanics and the homogenization theory of heterogeneous materials, a binary-medium constitutive model was established to reasonably express the stress-strain relationship of CIL under F-T conditions.

## 2. Materials and Methods

The tested loess was collected from the subgrade of the Wuzhong–Zhongwei intercity railway in northwestern China. The soil physical parameters were obtained through the gradation test and liquid-plastic limit test, as shown in Table 1.

The cement used in the CIL is Grade 42.5 ordinary Portland cement with a content of 6% and initial setting time of 4 h. The optimal moisture content of the CIL is 11.9%, and the maximum dry density is 1.85 g/cm^3^. The optimum moisture content was chosen to make samples in the tests, and the samples are cylinders with a height of 80 mm and a diameter of 60 mm. The sample preparation time was controlled within the initial setting time of the cement, and the curing time in the moisturizing cylinder at 20 °C is 14 days after the sample preparation is completed. The samples were then placed in an environment box for F-T cycles after curing and saturation. The triaxial compression tests were performed under consolidated undrained condition in a GDS triaxial testing system. The test parameters are listed in Table 2, and the test process is depicted in Figure 1.

## 3. Results and Discussion

### 3.1. Stress-Strain Curves of CIL

The results of triaxial tests using samples experienced F-T cycles at freezing temperatures of −5 °C, −10 °C, and −15 °C under different F-T cycle numbers (0, 1, 3, 6, 9, and 12) are as shown in Figure 2.

It can be seen that the stress-strain curves of CIL all show obvious strain-softening characteristics and brittleness properties, and the peak stress corresponds to a strain of 2% to 3%. This is due to the fact that the cement-hydrate will fill the pores between soil particles when the loess is mixed with cement and can cement the soil particles together, which significantly improves the bonding strength between soil particles [41].

Based on the stress-strain relationship of CIL, according to the damage mechanism of the natural structural soil and frozen soil with high-bond strength which is similar to CIL [42,43], the damage mechanism of CIL under shear loads is shown in Figure 3, and the curves can be divided into the following four stages.

(1)The linear-elastic stage. When the deviatoric stress is small, the cement hydrate is not damaged, and the stress-strain curve is almost elastic.(2)The elastic-plastic stage. As the axial strain gradually increases and the deviatoric stress reaches its peak, the effect of the bonding strength is enhanced and the friction between particles is weakened. The cement hydrate between the soil particles gradually breaks down, and the friction strength increases. Because the increased friction strength is much less great than the decreased bonding strength, the deformation resistance of the soil sample is weakened, and the slope of the curve gradually decreases to 0°.(3)The strain-softening stage. When the bonding strength is exceeded, the external load is fully borne by the friction strength, and the deviatoric stress decreases rapidly, showing a significant softening phenomenon.(4)The residual stability stage. After the occlusal and friction strength between particles is exceeded, the local shear fracture surface gradually penetrates, and the curve tends to a horizontal line. At this stage the corresponding deviatoric stress is the residual strength, that is, the sliding friction strength between the particles on the shear surface.

### 3.2. Effect of F-T Conditions on the Initial Tangent Modulus of CIL

The initial tangent modulus, $E_{0}$, reflects the initial deformation resistance of soils, which is closely related to the soil physical state, and is significantly influenced by the F-T conditions. It is calculated as follows:(1)$$E_{0}=\frac{{\left(\sigma_{1}-\sigma_{3}\right)}_{\varepsilon =0.2\%}}{\varepsilon_{0.2\%}}\times 100$$
where ${\left(\sigma_{1}-\sigma_{3}\right)}_{\varepsilon =0.2\%}$
is the deviatoric stress corresponding to the strain of 0.2%.

The relationship between the initial tangent modulus of CIL, the confining pressure and the factor of the F-T cycle number is shown in Figure 4.

It can be seen that the initial tangent modulus gradually increases with the growing confining pressures and decreases with the increase in the F-T cycle number, and the largest decreasing amplitude can be seen after the first F-T cycle. The initial tangent modulus also decreases with the reduction in the freezing temperature. For example, with the confining pressure of 150 kPa and the freezing temperature of −5 °C, the initial tangent modulus loses 10.6% of its original value after the first F-T cycle and 26.7% of its original value after the twelfth F-T cycle. At the freezing temperature of −10 °C with the same confining pressure, it loses 15.3% and 32.9%, respectively, after the first and twelfth F-T cycles. At the freezing temperature of −15 °C with the same confining pressure, it loses 16.7% and 46.4%, respectively, after the first and twelfth F-T cycles. This is due to the fact that the pore water undergoing volume expansion and contraction will alter the arrangement of soil particles and the connection between soil particles during the F-T process, and lead to the degradation of the occlusion and bonding strength. The lower the freezing temperature, the more intense the expansion and contraction of the pore water during the F-T cycles.

### 3.3. Effect of the F-T Conditions on the Strength of CIL

Based on the relationship between the deviatoric stress and axial strain of CIL samples, the peak stress of the strain-softening curves is taken as the critical strength (China National Standard, GB/T50123–2019) [44]. The relationship between the strength of CIL, the F-T cycle number and the factor of the confining pressure is shown in Figure 5.

From Figure 5, it can be seen that the F-T condition plays a significant role in affecting the strength of CIL. The strength decreases gradually with the increasing F-T cycle number, and the largest decreasing amplitude occurs after the first F-T cycle, while it gradually stabilizes after the sixth F-T cycle. The strength loss rate ζ is defined as the ratio of the test strength to the strength of the unfrozen sample at the same confining pressure, and the strength loss rates are listed in Table 3 under different conditions.

It can be seen from Table 3 that the strength loss rate ζ increases gradually with the growing F-T cycle number, but the increase rate gradually declines. For example, after the sixth F-T cycle at the freezing temperature of −5 °C, the strength loss rate ζ under 20 kPa has reached 32.92% compared to 34.79% after the twelfth F-T cycle at the same freezing temperature and confining pressure. The reason is that the F-T cycles before the sixth cycle substantially change the internal structure (particle arrangement and connection) of CIL and generate many macropores; thus, the freezing and thawing process of the water after the sixth F-T cycle in the existing macropores slightly changes the CIL structure. The lower the freezing temperature, the larger the strength loss rate. However, the effect of the freezing temperature on the strength loss rate gradually decreases with the growing F-T cycle number.

The confining pressures also have a significant effect on the strength loss rate. The higher the confining pressure, the stronger the support effect on the numerous fragile macropores formed during the F-T cycles, and the smaller the strength loss rate. For example, at the freezing temperature of −5 °C, the values of strength loss rate are 34.79%, 18.59% and 13.50% after the twelfth F-T cycle at the confining pressure of 20 kPa, 60 kPa and 150 kPa, respectively.

### 3.4. Nonlinear Mohr-Coulomb Strength Criterion

The strength of CIL shows a nonlinear relationship with the confining pressure (Figure 5); thus, the Weibull function is used to express the relation between the axial pressure *σ*_1_ and the confining pressure *σ*_3_ when the sample is damaged [45,46].
(2)$$\sigma_{1}=\sigma_{0}+a_{s}p_{a}\left\{1-\exp [-{\left(\frac{\sigma_{3}}{b_{s}p_{a}}\right)}^{n_{s}}]\right\}$$
where $\sigma_{0}$
is the unconfined compressive strength, $p_{a}$
is the standard atmospheric pressure (*p*_a_ = 101.33 kPa), and $a_{s}$, $b_{s}$
and $n_{s}$
are the material parameters. Those parameters can be obtained through a calculation. The calculated curves for the relationship between *σ**_1_* and *σ**_3_* are shown in Figure 6.

Based on the Mohr-Coulomb theory, the function of the Mohr circle can be expressed as:(3)$$f\left(\sigma ,\tau ,\sigma_{1},\sigma_{3}\right)={\left(\sigma -\frac{\sigma_{1}+\sigma_{3}}{2}\right)}^{2}+\tau^{2}-{\left(\frac{\sigma_{1}-\sigma_{3}}{2}\right)}^{2}=0$$

Equation (2) can be rewritten as:(4)$$g\left(\sigma_{1},\sigma_{3}\right)=\sigma_{1}-\sigma_{0}-a_{s}p_{a}\left\{1-\exp [-{\left(\frac{\sigma_{3}}{b_{s}p_{a}}\right)}^{n_{s}}]\right\}=0$$

From the envelope theory, the following function can be obtained.
(5)$$\frac{\partial f\left(\sigma ,\tau ,\sigma_{1},\sigma_{3}\right)}{\partial \sigma_{1}}\cdot \frac{\partial g\left(\sigma_{1},\sigma_{3}\right)}{\partial \sigma_{3}}-\frac{\partial f\left(\sigma ,\tau ,\sigma_{1},\sigma_{3}\right)}{\partial \sigma_{3}}\cdot \frac{\partial g\left(\sigma_{1},\sigma_{3}\right)}{\partial \sigma_{1}}=0$$

Differentiating Equations (3) and (4), respectively, the following functions are obtained.
(6)$$\frac{\partial f}{\partial \sigma_{1}}=\sigma_{3}-\sigma$$
(7)$$\frac{\partial f}{\partial \sigma_{3}}=\sigma_{1}-\sigma$$
(8)$$\frac{\partial g}{\partial \sigma_{1}}=1$$
(9)$$\frac{\partial g}{\partial \sigma_{3}}=-a_{s}n_{s}p_{a}\frac{\sigma_{3}^{n_{s}-1}}{b_{s}^{n_{s}}p_{a}^{n_{s}}}\exp [-{\left(\frac{\sigma_{3}}{b_{s}p_{a}}\right)}^{n_{s}}]$$

By substituting Equations (6)–(9) into Equation (5), the expression for the normal stress on the shear surface is obtained as follows.
(10)$$\sigma =\frac{\sigma_{1}+a_{s}n_{s}p_{a}{\left(\sigma_{3}/b_{s}p_{a}\right)}^{n_{s}}\exp [-{\left(\sigma_{3}/b_{s}p_{a}\right)}^{n_{s}}]}{1+a_{s}n_{s}p_{a}\left(\sigma_{3}^{n_{s}-1}/b_{s}^{n_{s}}p_{a}^{n_{s}}\right)\exp [-{\left(\sigma_{3}/b_{s}p_{a}\right)}^{n_{s}}]}$$

The expression for the shear stress on the shear surface can then be obtained.
(11)$$\tau =\sqrt{{\left(\frac{\sigma_{1}-\sigma_{3}}{2}\right)}^{2}-{\left\{\frac{\sigma_{1}+a_{s}n_{s}p_{a}{\left(\sigma_{3}/b_{s}p_{a}\right)}^{n_{s}}exp [-{\left(\sigma_{3}/b_{s}p_{a}\right)}^{n_{s}}]}{1+a_{s}n_{s}p_{a}\left(\sigma_{3}^{n_{s}-1}/b_{s}^{n_{s}}p_{a}^{n_{s}}\right)exp [-{\left(\sigma_{3}/b_{s}p_{a}\right)}^{n_{s}}]}-\frac{\sigma_{1}+\sigma_{3}}{2}\right\}}^{2}}\;$$

Equations (6), (9) and (11) are the expressions of the nonlinear Mohr-Coulomb strength criterion for CIL, and the Mohr circle and its strength envelope were plotted, as shown in Figure 7.

## 4. The Binary-Medium Model of CIL

According to the basic properties, deformation characteristics, and research methods for geotechnical materials, Shen [47,48] established a theoretical framework of geotechnical damage mechanics, proposed a binary-medium model, and abstracted the geotechnical materials into binary-medium materials comprised of structural elements (such as structural blocks and structural belts). At present, the binary-medium model has been widely applied and verified in geotechnical mechanics [49,50,51].

The binary-medium model comprises bonded elements and frictional elements in parallel, as shown in Figure 8.

The bonded element is simulated in series as a brittle bond and a spring, and the frictional element is simulated in series as a plastic slider and a spring. When the strain is small, the bonded elements will come into play. When the strain enlarges, the load on the frictional element gradually increases until the cement rod on the cemented components breaks and the frictional element begins to bear the external load. The stress-strain curves of the bonded and frictional elements are also shown in Figure 8. The sum of the superimposed stress-strain curves of the bonded and the frictional element is the total stress-strain curve of the soil, in which the first increase and then decrease in the stress of the bonded elements and the increase in the stress of the frictional elements reflect their conversion relationship.

As a kind of reinforced composite material, the strength of CIL is mainly comprised of the cementation strength of the cement hydrate and the friction strength between particles, and its loading mechanism is in accordance with the binary-medium theory under the theoretical framework of geotechnical breakage mechanics. According to the binary-medium theory, the CIL is abstractly divided into bonded elements and frictional elements, both of which share the external load. The deformation modulus of the bonded element is much higher than that of the frictional element. The bonded elements are gradually broken and transformed into frictional elements in the shearing process, and the initial micro fractures in the soil are gradually expanded, and form a local shear zone along with the damage of the bonded element. When the strain further increases, the local shear zone will develop into the macroscopic rupture surface and expand in the whole soil body, finally leading to the destruction of the soil structure.

### 4.1. Stress-Strain Characteristics of Bonded and Frictional Elements

When the cement is added into the loess, the cement hydrate and soil particles are cemented to form the bonded elements, providing a strong cohesive force. In the idealized model, most of the external load is borne by the bonded elements when the strain is small, and the stress-strain curve of the bonded element is considered to be linearly elastic. The stress-strain relationship of the bonded element can be expressed as follows [45]:(12)$$\sigma_{b}=E_{b}\varepsilon_{b}\;$$
where $\sigma_{b}$
is the stress of the bonded element, $E_{b}$
is the initial modulus of the bonded element, whose value can be taken as the initial tangent modulus of the soil in the idealized model, and $\varepsilon_{b}$
is the strain of the bonded element.

With the raising strain, when it exceeds the linear elastic range, the bonded elements start to be damaged and transformed into the frictional elements. The bonded and frictional elements bear the load together. With the further growth in the strain, more bonded elements are damaged and transformed into frictional elements; therefore, the number of frictional elements and the stress borne by frictional elements will gradually increase. Since the deformation modulus of the bonded elements is much larger than that of the frictional elements, the slope of the stress-strain curve at this time shows a gradual decreasing trend. When the strain reaches and exceeds the strain corresponding to the peak stress, the bonding strength produced by the cement hydrate decreases rapidly and falls to zero, and the deviatoric stress decreases close to the residual strength. The bonded elements are considered to be completely damaged and transformed into frictional elements, and the frictional elements bear all the external load. The Duncan-Chang hyperbolic model is adopted to represent the stress-strain curve of the frictional element, and the expression is as follows [45]:(13)$$\sigma_{f}=\frac{\varepsilon_{f}}{a+b\varepsilon_{f}}\;$$
where $\sigma_{f}$
is the stress of the frictional element, $\varepsilon_{f}$
is the strain of the frictional element, and a
and b
are material parameters.

### 4.2. Binary-Media Model of CIL

In accordance with the homogenization theory of heterogeneous materials, a representative elementary volume (REV) is taken out of the soil, and its average stress can be expressed as follow [48]:(14)$$\sigma_{ij}=\frac{1}{V}\int \sigma_{ij}^{loc}dV=\frac{1}{V}\int \sigma_{ij}^{loc}d\left(V_{b}+V_{f}\right)=\frac{1}{V}\int \sigma_{ij}^{loc}dV_{b}+\frac{1}{V}\int \sigma_{ij}^{loc}dV_{f}\;$$
where V
is the volume of the REV, $V_{b}$
and $V_{f}$
are the volume of the bonded elements and the volume of the frictional elements, respectively, and $\sigma_{ij}$
and $\sigma_{ij}^{loc}\;$
are the average stress of REV and local stress of the REV, respectively.

The stress of the bonded elements and the stress of the frictional elements can be expressed as follows [48]:(15)$$\sigma_{ij}^{b}=\frac{1}{V_{b}}\int \sigma_{ij}^{loc}dV_{b}$$
(16)$$\sigma_{ij}^{f}=\frac{1}{V_{f}}\int \sigma_{ij}^{loc}dV_{f}$$
where $\sigma_{ij}^{b}$
and $\sigma_{ij}^{f}$
are the local stress of the bonded elements and the local stress of the frictional elements, respectively.

By substituting Equations (15) and (16) into Equation (14), the average stress of the REV can be obtained.
(17)$$\sigma_{ij}=\frac{V_{b}}{V}\frac{1}{V_{b}}\int \sigma_{ij}^{loc}dV_{b}+\frac{V_{f}}{V}\frac{1}{V_{f}}\int \sigma_{ij}^{loc}dV_{f}=\frac{V_{b}}{V}\sigma_{ij}^{b}+\frac{V_{f}}{V}\sigma_{ij}^{f}\;$$

The average strain of the REV can be obtained by using the similar procedure.
(18)$$\varepsilon_{ij}=\frac{V_{b}}{V}\varepsilon_{ij}^{b}+\frac{V_{f}}{V}\varepsilon_{ij}^{f}\;$$
where $\varepsilon_{ij}$
is the average strain of the REV, $\varepsilon_{ij}^{b}$
and $\varepsilon_{ij}^{f}$
are the local stresses of the bonded elements and frictional elements, respectively. The volume damage rate *λ* is defined as the ratio of the volume of the frictional elements in the REV to the volume of the REV.
(19)$$\lambda =\frac{V_{f}}{V}$$

Equation (19) is substituted into Equations (17) and (18) to derive the following functions.
(20)$$\sigma_{ij}=\left(1-\lambda \right)\sigma_{ij}^{b}+\lambda \sigma_{ij}^{f}$$
(21)$$\varepsilon_{ij}=\left(1-\lambda \right)\varepsilon_{ij}^{b}+\lambda \varepsilon_{ij}^{f}$$

By using the linear-elastic model to represent the stress-strain relationship of the bonded element and the Duncan-Chang hyperbolic model to simulate the stress-strain relationship of the frictional element under a triaxial stress condition, Equation (20) can be simplified to the following function.
(22)$$\sigma_{d}=\left(1-\lambda \right)E_{b}\varepsilon_{d}^{b}+\lambda \frac{\varepsilon_{d}^{f}}{a+b\varepsilon_{d}^{f}}\;$$
where $\sigma_{d}$
is the average stress of the REV, $\varepsilon_{d}^{b}$
and $\varepsilon_{d}^{f}$
are the local strain of the bonded elements and the local strain of the frictional elements, respectively.

By using the local strain coefficient *c* to describe the non-uniform deformation of the bonded and frictional elements, the relationship between the strain in the frictional element $\varepsilon_{d}^{f}$
and the strain in the RVE $\varepsilon_{d}$
can be established:(23)$$\varepsilon_{d}^{f}=c\varepsilon_{d}$$

By substituting Equation (23) into Equation (21), the relationship between the strain in the bonded elements and the strain in the RVE can be derived.
(24)$$\varepsilon_{d}^{b}=\frac{\left(1-c\lambda \right)}{\left(1-\lambda \right)}\varepsilon_{d}\;$$

By substituting Equations (23) and (24) into Equation (22), the following function can be obtained.
(25)$$\sigma_{d}=\left(1-c\lambda \right)E_{b}\varepsilon_{d}+\lambda \frac{c\varepsilon_{d}}{a+bc\varepsilon_{d}}\;$$

By comparison, the Weibull function is used to simulate the breaking and conversion process of the bonded elements to the frictional elements [45,46].
(26)$$\lambda =1-exp [-\alpha {\left(\frac{\varepsilon_{d}}{i}\right)}^{n}]$$
where i, α
and n
are material parameters.

By substituting Equation (26) into Equation (25), the stress-strain expression of CIL under the triaxial shear condition can be obtained.
(27)$$\sigma_{d}=\left\{1-c+cexp [-\alpha {\left(\frac{\varepsilon_{d}}{i}\right)}^{n}]\right\}\cdot E_{b}\varepsilon_{d}+\left\{1-exp [-\alpha {\left(\frac{\varepsilon_{d}}{i}\right)}^{n}]\right\}\frac{c\varepsilon_{d}}{a+bc\varepsilon_{d}}\;$$

### 4.3. Model Validation

Combining the test results with Equation (27), the model calculation results can be obtained, as shown in Figure 9 and Table 4. It can be seen that the model can greatly simulate the stress-strain relationship of CIL, especially for the strain-softening phenomenon.

Combining the calculation results with Equation (26), the values of the breakage rate used in the present constitutive model can be obtained, and Figure 10 shows the variation in breakage rate versus strain under different conditions.

It can be seen that the value of the breakage rate tends to zero when the strain is small, which corresponds to the fact that the external load in the linear-elastic phase of the test is mainly borne by the bonded element and there is no transition from the bonded element to the frictional element. As the strain increases, the breakage rate rises rapidly from 0 to 1 after the inflection point, and a large number of bonded elements convert to the frictional elements. This is the elastic-plastic phase and the strain-softening phase where the stress-strain curve moves beyond the peak and decreases rapidly. When the strain continues to increase, the breakage rate reaches and remains constant at 1, and the stress-strain curve enters the residual strength phase. This is also in line with the significance of the breakage rate, which shows that it can simulate the transition process from the bonded elements to the frictional elements well.

## 5. Conclusions

Through the triaxial tests under various conditions, the change in the deformation and strength of CIL were investigated, and a binary-medium model was established to describe the deformation behavior of CIL. The main conclusions are as follows:(1)The stress-strain curves of CIL show an obvious strain-softening phenomenon. The initial tangent modulus and strength of CIL increase with the growth in confining pressure and gradually decrease with the increasing F-T cycle number and the reduction in the freezing temperature. They all lose at least one-third of their original values after the twelfth F-T cycle with the confining pressure of 150 kPa and the freezing temperature of −15 °C.(2)To reflect the relationship between the shear strength and normal stress of CIL, a nonlinear Mohr-Coulomb strength criterion was proposed by adopting the Weibull function to describe the relationship between the axial stress and confining pressure when the soil sample failure.(3)Based on the damage mechanism of CIL and the homogenization theory of the heterogeneous materials, a binary-medium constitutive model was established. The model can not only describe the entire damage process of CIL, but also predict exactly the residual strength under F-T conditions.

## Figures and Tables

**Figure 1 materials-15-07042-f001:**
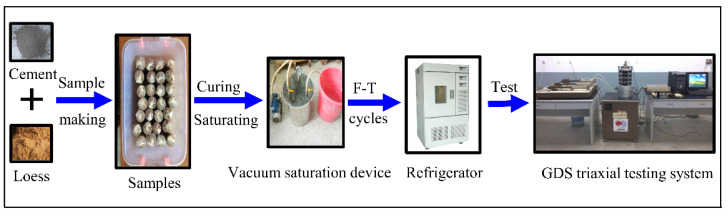
Schematic diagram of experiment process.

**Figure 2 materials-15-07042-f002:**
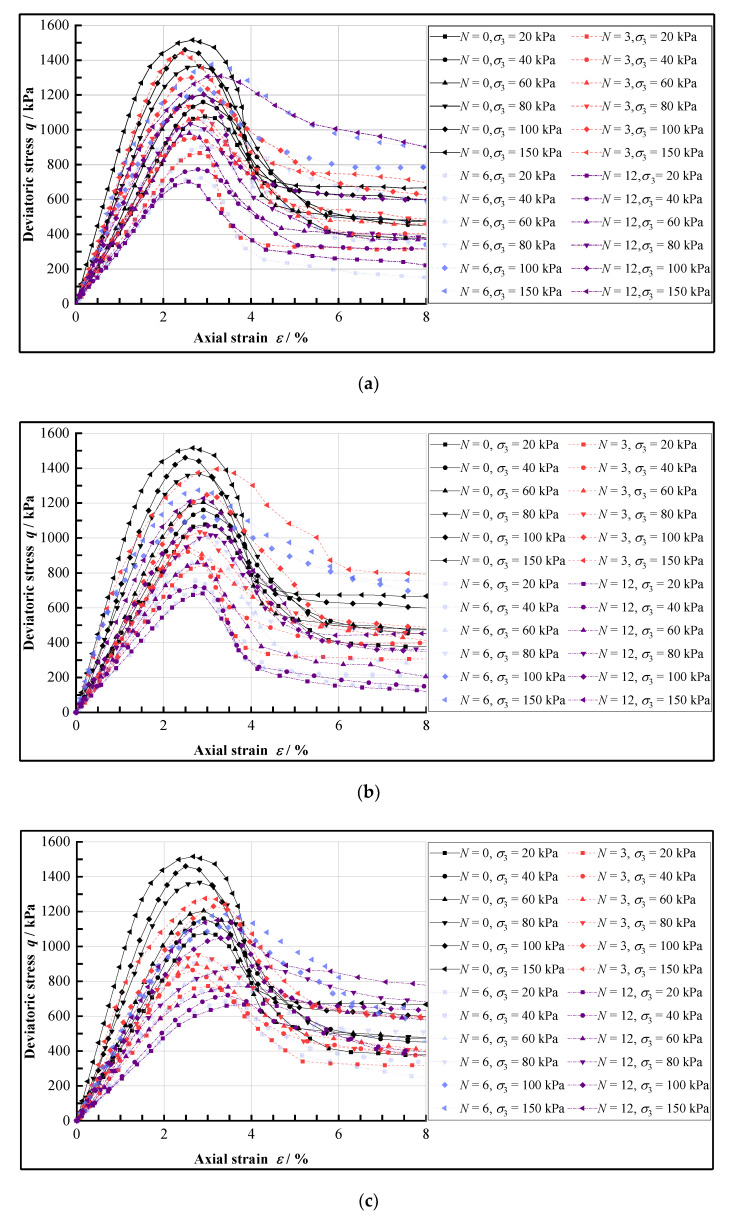
The stress-strain curves of CIL under different freezing temperatures, (**a**) The freezing condition of −5 °C, (**b**) The freezing condition of −10 °C, (**c**) The freezing condition of −15 °C.

**Figure 3 materials-15-07042-f003:**
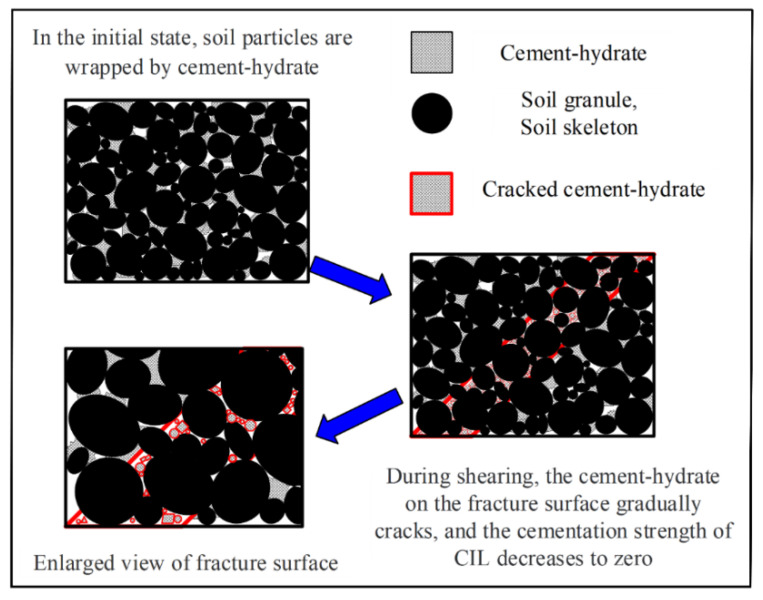
The damage mechanism of the CIL sample.

**Figure 4 materials-15-07042-f004:**
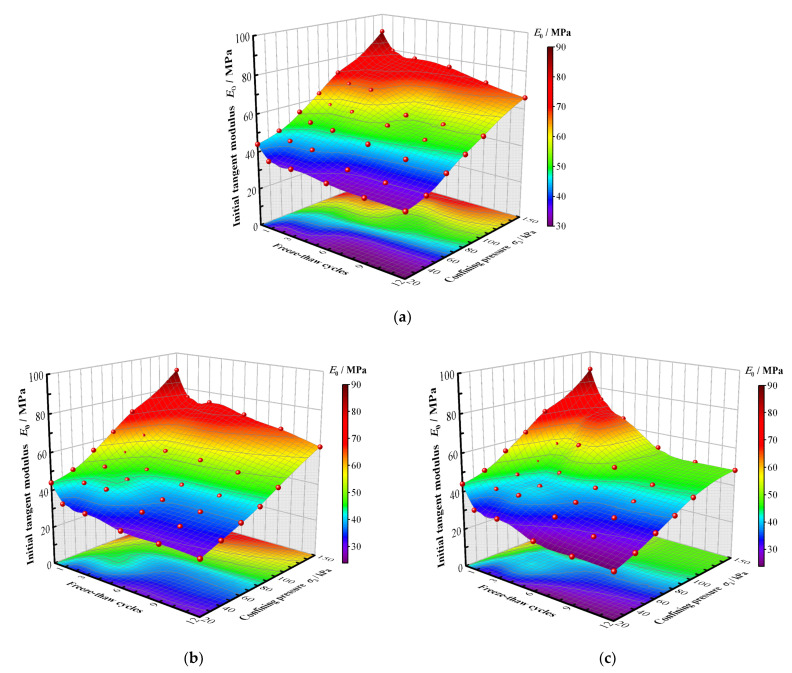
The initial tangent moduli of CIL under different F-T cycles and confining pressures, (**a**) The freezing condition of −5 °C, (**b**) The freezing condition of −10 °C, (**c**) The freezing condition of −15 °C.

**Figure 5 materials-15-07042-f005:**
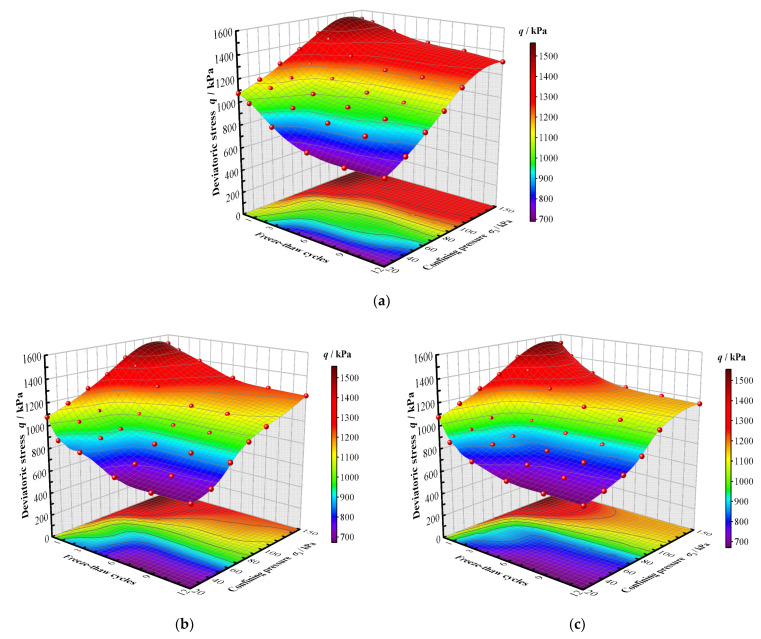
The relationship between strength and confining pressure under different F-T cycles, (**a**) The freezing condition of −5 °C, (**b**) The freezing condition of −10 °C, (**c**) The freezing condition of −15 °C.

**Figure 6 materials-15-07042-f006:**
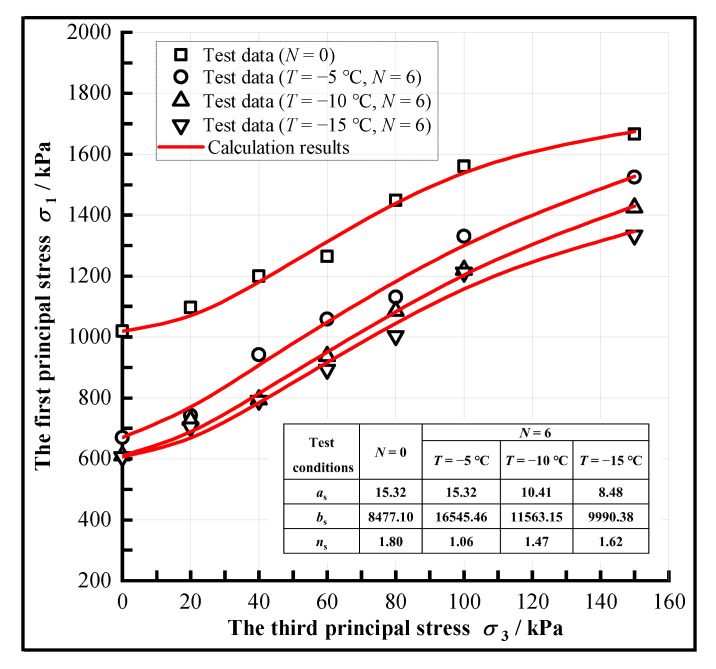
The test and calculation results of *σ*_1_ vs. *σ*_3_ of CIL.

**Figure 7 materials-15-07042-f007:**
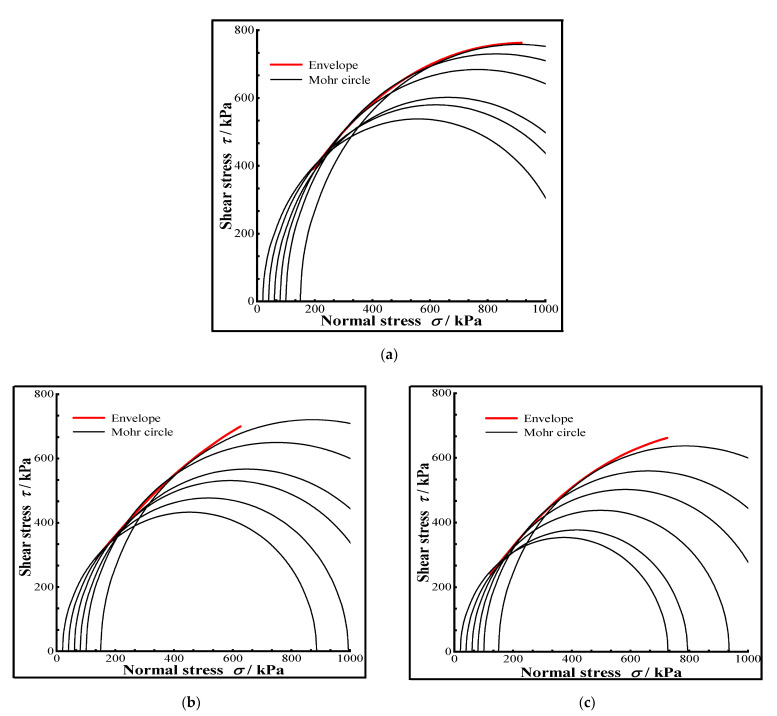
The strength envelopes of CIL under different conditions, (**a**) The condition without F-T cycle, (**b**) The freezing condition of −5 °C and the tenth F-T cycle, (**c**) The freezing condition of −10 °C and the sixth F-T cycle.

**Figure 8 materials-15-07042-f008:**
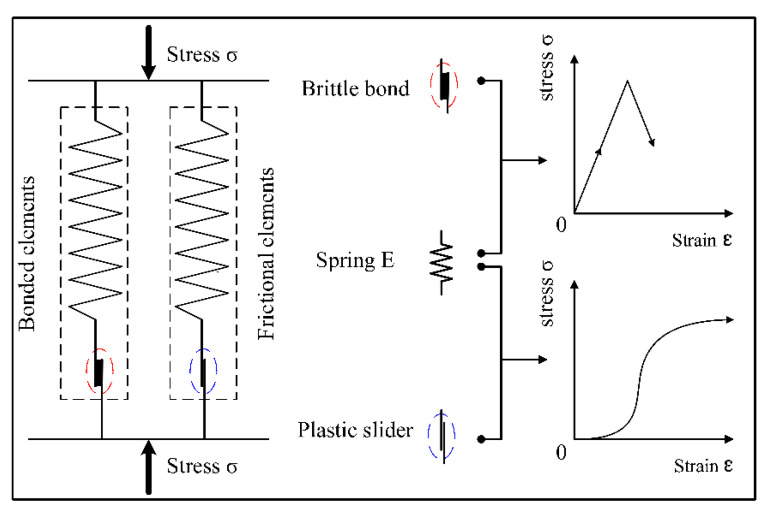
The binary-medium model component diagram.

**Figure 9 materials-15-07042-f009:**
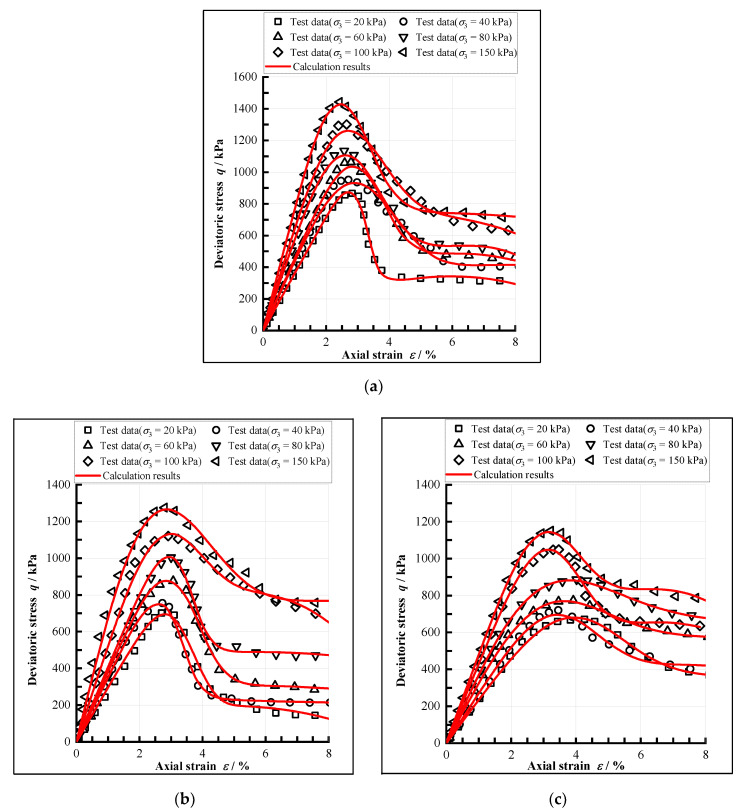
Test and calculation results after the twelfth F-T cycle under different freezing temperatures, (**a**) The freezing temperature is −5 °C, (**b**) The freezing temperature is −10 °C, (**c**) The freezing temperature is −15 °C.

**Figure 10 materials-15-07042-f010:**
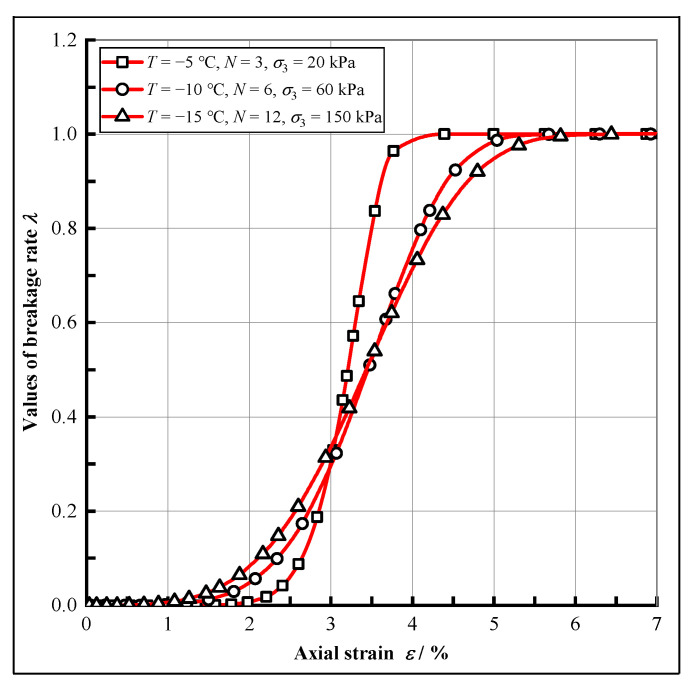
The changes law of the breakage rate.

**Table 1 materials-15-07042-t001:** The physical parameters of the tested loess.

Plastic Limit/%	Liquid Limit/%	Particle-Size (mm) Distribution/%					
		<0.005	0.005–0.05	0.05–0.075	0.075–0.1	0.1–0.25	>0.25
13.5	21.7	9.28	63.11	18.02	6.56	3.03	0

**Table 2 materials-15-07042-t002:** Parameters for triaxial test.

Freezing Temperature/°C	Freezing Time/h	Melting Temperature/°C	Melting Time/h	Test Rate /mm/min	Confining Pressure/kPa	Number of F-T Cycles
−5, −10, −15	24	15	24	0.5	20, 40, 60 80, 100, 150	0, 1, 3 6, 9, 12

**Table 3 materials-15-07042-t003:** The strength loss rates under different conditions.

Test Conditions	T = −5 °C *N* = 3	T = −5 °C *N* = 6	T = −5 °C *N* = 12	T = −10 °C *N* = 3	T = −10 °C *N* = 6	T = −10 °C *N* = 12	T = −15 °C *N* = 3	T = −15 °C *N* = 6	T = −15 °C *N* = 12
*σ*_3_ = 20 kPa	19.6%	32.9%	34.8%	21.5%	34.3%	36.7%	28.3%	36.5%	37.7%
*σ*_3_ = 40 kPa	17.7%	22.2%	33.4%	20.6%	35.0%	37.7%	25.5%	35.2%	37.9%
*σ*_3_ = 60 kPa	11.7%	17.1%	18.6%	21.4%	27.2%	28.8%	25.4%	30.8%	35.9%
*σ*_3_ = 100 kPa	10.9%	15.7%	17.6%	14.6%	23.3%	26.7%	15.7%	23.8%	28.2%
*σ*_3_ = 150 kPa	4.9%	9.3%	13.5%	7.9%	16.0%	19.1%	15.8%	21.9%	23.8%

**Table 4 materials-15-07042-t004:** The parameters of the model.

Test Conditions	*c*	*α*	*i*	*n*	*a*	*b*	*E**_b_*/MPa
T = −5 °C, *N* = 3, *σ*_3_ = 40 kPa	0.11	4.36	6.03	3.27	−0.0004	0.000057	44.1
T = −5 °C, *N* = 3, *σ*_3_ = 80 kPa	0.69	2.77	4.62	3.39	−0.0134	0.0013	58.3
T = −5 °C, *N* = 3, *σ*_3_ = 150 kPa	1.04	7.00	5.29	3.53	0.0012	0.00089	77.0
T = −10 °C, *N* = 6, *σ*_3_ = 40 kPa	1.01	2.09	3.71	6.02	−0.0013	0.0044	34.9
T = −10 °C, *N* = 6, *σ*_3_ = 80 kPa	1.09	2.44	4.14	6.51	0.0027	0.001	41.7
T = −10 °C, *N* = 6, *σ*_3_ = 150 kPa	0.58	0.05	1.01	2.33	−0.0042	0.00027	70.1
T = −15 °C, *N* = 12, *σ*_3_ = 40 kPa	1.11	5.15	6.19	3.56	0.0018	0.0013	27.1
T = −15 °C, *N* = 12, *σ*_3_ = 80 kPa	1.02	1.58	4.74	1.96	0.0014	0.0012	37.4
T = −15 °C, *N* = 12, *σ*_3_ = 150 kPa	−0.13	2.94	4.99	3.88	0.0004	0.00011	48.2

## Data Availability

The data presented in this study are available on reasonable request from the corresponding author.

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
