# Peer review of "Mechanical Properties and Constitutive Model of the Cement-Improved Loess under Freeze-Thaw Conditions"

_materials, 2022, doi:10.3390/ma15197042_

Round 1

Reviewer 1 Report

Mechanical properties and constitutive model of the cement improved loess under freeze–thaw conditions

The article is well written. Few suggestions are recommeded.

The Objectives and aims are missing in the abstract. Abstract need revision with some quantitative results.

Some more latest studies are required in the introduction section to further highlight the importance of this study.

Khurram, N., Khan, K., Saleem, M. U., Amin, M. N., & Akmal, U. (2018). Effect of elevated temperatures on mortar with naturally occurring volcanic ash and its blend with electric arc furnace slag. Advances in Materials Science and Engineering, 2018.

Khan, M. I., Sutanto, M. H., Napiah, M. B., Khan, K., & Rafiq, W. (2021). Design optimization and statistical modeling of cementitious grout containing irradiated plastic waste and silica fume using response surface methodology. Construction and Building Materials, 271, 121504.

Table 1. The physical parameters of the tested loess. How properties were determined.

Equations must be quoted with proper references.

List of abbreviations and symbols are missing.

Authors must summarized results in more systematic way with reference to the previous studies.

Also, Conclusions are too limited to proof the significant outcome of this study.

Author Response

Thank you very much for your comments and suggestions. We have made some revisions and statements for some problems according to the reviewing comments in our revision manuscript, marked by blue color. Our response to the comments is given in the attachment. Please see the attachment.

Reviewer 2 Report

The theme of the article is importance and attracts attention as well. However, this research paper is more a work in progress than a finished product. Unfortunately, it does not conform to the journal requirement at this stage. It is necessary to overhaul editing and resubmit.

It can be briefly summarized as follows.

1. why did you not evaluate XRD and MIP in this paper? It is the basic test for evaluating F-T, and it hardly explanate the mechanism without XRD and MIP data. This data is necessary supplementary.

2. for figure 3. How to confirm the damage mechanism is correct? Are there any data to support this? Such as SEM?

3. it is necessary to modify more details of the theoretical basis for test results and the author’s explanations in this paper. Furthermore, please elaborate on your originality in this paper.

Author Response

Thank you very much for your comments. We have carefully made revisions according to your suggestions. All changes are marked by blue characters in this version. Our response to the key points is in the  attachment. Please see the attachment.

Reviewer 3 Report

The work is interesting but it looks to me that the background study was not investigated well. There are many studies in this area. In fact, the design of pavement using mechanistic method is the key research area at present. This study has some new finding. But it is impossible to understand the motivation and improvement with comparison to the existing knowledge. Please discuss the literature more elaborately using more examples. Most, if not all, of the references used are local to authors. Please use some international references. One example, Effects of Large Freeze-Thaw Cycles on Stiffness and Tensile Strength of Asphalt Concrete. ASCE Journal of Cold Regions Engineering  The conclusion section can be rewritten to concisely give the conclusion. In current state, this section became a discussion section. Or, a new conclusion section can be added.

Author Response

(The authors gave the same response as above.)

Round 2

Reviewer 3 Report

Thanks for revising the article.